# Attention-Enhanced Unpaired xAI-GANs for Transformation of Histological Stain Images

**DOI:** 10.3390/jimaging10020032

**Published:** 2024-01-25

**Authors:** Tibor Sloboda, Lukáš Hudec, Matej Halinkovič, Wanda Benesova

**Affiliations:** Faculty of Informatics and Information Technology, Slovak Technical University, Ilkovičova 2, 842 16 Bratislava, Slovakia; lukas.hudec@stuba.sk (L.H.); matej.halinkovic@stuba.sk (M.H.); vanda_benesova@stuba.sk (W.B.)

**Keywords:** CycleGAN, histology, breast cancer, p63, attention, generation

## Abstract

Histological staining is the primary method for confirming cancer diagnoses, but certain types, such as p63 staining, can be expensive and potentially damaging to tissues. In our research, we innovate by generating p63-stained images from H&E-stained slides for metaplastic breast cancer. This is a crucial development, considering the high costs and tissue risks associated with direct p63 staining. Our approach employs an advanced CycleGAN architecture, xAI-CycleGAN, enhanced with context-based loss to maintain structural integrity. The inclusion of convolutional attention in our model distinguishes between structural and color details more effectively, thus significantly enhancing the visual quality of the results. This approach shows a marked improvement over the base xAI-CycleGAN and standard CycleGAN models, offering the benefits of a more compact network and faster training even with the inclusion of attention.

## 1. Introduction

### 1.1. Background

Cancer is the leading cause of disease-related death globally [1,2], emphasizing the need for effective diagnostic tools. Histological analysis, specifically Hematoxylin and Eosin (H&E) staining, plays a crucial role in cancer detection [3]. In certain cancers, like metaplastic breast cancer, the p63 immunohistochemical stain becomes vital for identifying myoepithelial cell invasion, a marker of cancer [4]. The combination of H&E and p63 stains has been shown to enhance the accuracy of cancer diagnoses, including in breast and laryngeal cancers [5,6], aiding in differentiating invasive carcinoma from benign lesions.

### 1.2. Challenge

Despite its utility, the p63 stain presents significant challenges. Its high cost, compared to H&E, higher difficulty of preparation and additional time required after H&E staining [7] make it a less-than-ideal solution. Additionally, the process of aligning and comparing traditional H&E stained slides with p63 stained slides is fraught with difficulties due to inherent warping issues and differences between adjacent slices.

### 1.3. Our Contribution Using Enhanced xAI-CycleGAN Architecture

Our modified xAI-CycleGAN, based on CycleGAN [8], includes convolutional attention and a context-based loss function. These upgrades help it distinguish between structural and color aspects, enhancing visual quality while keeping the network architecture compact and training time short. We have additionally improved the editing capabilities as compared to our previous method [9] to make it more predictable and visually interpolable in a controlled manner, although not yet usable in a clinical setting.

## 2. Related Work

Various works have demonstrated relatively successful virtual staining of histopathological tissue [10] and transformation from one type of stained tissue to another [10,11,12]. Registration remains a consistent issue due to the nature of the problem, though it is possible. For paired transformation, various GAN architectures were used, from a simple GAN and DCGAN [13] as well as sequential cascaded GANs which show common use in the medical domain [14].

For the unpaired approach, CycleGAN [8] seems to be the dominant approach in nearly all cases. Therefore, we will look a bit closer at some related work that we predominantly use or relate to below.

### 2.1. Overview of xAI-CycleGAN

xAI-CycleGAN is an enhanced version of the CycleGAN [8] architecture that aims to improve the convergence rate and image quality in unsupervised image-to-image transformation tasks [15]. It incorporates the concepts of explainability to provide a more powerful and versatile generative model. One of the primary contributions of xAI-CycleGAN is the incorporation of explainability-driven training. Inspired by the work of Nagisetty et al. [16], xAI-CycleGAN utilizes saliency maps from the discriminator to mask the gradients of the generator during back-propagation. Saliency maps provide the generator with more effective feedback as they help focus the model’s learning on features that are more important for fooling the discriminator.

In addition, xAI-CycleGAN leverages the insights from the work of Wang M.’s Mask CycleGAN [17], which introduces an interpretable latent variable using hard masks on the input. These interpretable masks help the generator respect the effects of the mask and prevent information from leaking and influencing the generator. By combining these approaches, xAI-CycleGAN achieves enhanced explainability and convergence by taking advantage of information leakage into the interpretable latent variable. One issue with this architecture is the production of counterfactuals by the generator and various artifacts and repeating patterns, which may be worsened in the histopathological domain and must be addressed.

We do this by introducing a new loss that attempts to preserve the structure and enforce context by separating the context and style, forcing all style transformations to happen at the latent level of the generative model and preventing the encoder or decoder portion from doing any color transformation.

### 2.2. cCGAN for Stain Transformation

cCGAN (conditional CycleGAN) [18] is a notable approach in the domain of stain transformation for histology images. It aims to translate images from one stain to another while preserving crucial structural information.

The cCGAN approach employs a conditional variant of the CycleGAN architecture, incorporating additional information, such as staining information, as input to guide the translation process.

While cCGAN has shown promise in transforming between stains, there are some limitations to consider. One challenge is the preservation of fine-grained structural details during the transformation process. Due to the absence of explicit constraints on structure preservation, cCGAN may struggle to preserve important structural information accurately, as shown in their results.

We address the issue of structure preservation by incorporating context loss, which enforces consistency between encoded representations and emphasizes the conservation of structural features during the transformation.

### 2.3. Transformer and Cycle-Consistent GAN

Recent experiments [19] have demonstrated the potential benefits of combining vision transformers [20] with the standard CycleGAN architecture. Replacing the CNN encoder with an attention-based alternative produces qualitatively and quantitatively superior results while managing to converge faster.

The vision transformer relies on the standard self-attention [21], which has quadratic computational complexity in relation to the number of patches used to represent the input image. This problem is severe, especially when dealing with high-resolution images such as WSIs. The availability of hardware thus limits the potential for improvements.

Other attention methods are better suited for tasks where a perfect global understanding of the input image is not necessary. This is why our approach utilizes convolutional attention with iterative filter reduction based on the depth in the auto-encoder architecture. Additionally, we apply attention in the decoder stage rather than the encoder stage and control context using our context-preserving loss, which we believe has similar potential and effects but is more accessible to train and requires less memory.

### 2.4. Advancements in Editable Outputs Using Modified SeFa Algorithm

The development of the Closed-Form Factorization of Latent Semantics in GANs (SeFa) algorithm marked a significant advancement in the field of generative networks. Introduced by Zhou et al., this method offers an innovative way to semantically manipulate generated outputs, such as altering expressions, colors and postures [22]. The SeFa algorithm achieves this by identifying and utilizing high variability vectors within the latent space, which are then applied to the network layers through eigenvalue multiplication.

Other image modification techniques exist, such as sequential GANs and inpainting methods [23,24,25]. Many of these also utilize variations of the SeFa principle, with some incorporating attention-based mechanisms for enhanced performance [21].

However, these techniques are primarily effective with simpler generative models that rely on noise distributions for image creation. In more complex models like CycleGAN, which utilizes a four-dimensional latent weight matrix in its auto-encoder-like structure, the direct application of the SeFa method is not feasible.

To overcome this limitation, we have adapted the SeFa approach for use in xAI-CycleGAN in our previous work [9]. This modification involves employing a singular layer with an interpretable latent variable, allowing for some level of semantically meaningful modifications. While this method diverges from the original SeFa in terms of losing certain semantically interpretable aspects, it still opens the possibility for image editing in the CycleGAN architecture.

In this paper, we demonstrate further improvements to the approach, which features significant improvements to the interpretability of the editable parameters in terms of stable and predictable changes in the resulting images.

## 3. Dataset

We use a double-staining dataset consisting of the H&E and p63 tissue stains. This is a private dataset obtained from the Institute for Clinical and Experimental Medicine [26] specifically for the purpose of this work. The data are fully anonymized and do not reveal any identifying patient information. The choice of the p63 IHC diagnostic stain along with its relation to the cancer type were not chosen for the purpose of clinical diagnostics but purely for the purpose of researching advances in computer vision approaches for virtual staining.

H&E is commonly used to highlight various types of tissue and cells with varying hues from light pink to dark purple. Cell nuclei are usually highlighted with a dark purple, allowing pathologists to differentiate them from the surrounding tissue.

While H&E alone has sufficient diagnostic information to identify the presence of cancer, it is a bit more challenging and slow to identify metaplastic breast cancer signatures and presence from it. For this reason, p63 is also used. This immunohistochemical stain targets a specific protein present within myoepithelial cells surrounding ducts in the breast tissue, which allows pathologists to identify these cells easily. When these cells are not in a particular arrangement or infiltrate the tissue, it is likely due to them being cancerous in nature.

Preparation with p63 can be lengthy, has increased complexity and is expensive compared to H&E as well [7], which gives us the motivation to attempt to acquire p63 from the cheap H&E stain as it would take significantly shorter to only prepare an H&E stain and utilize digital re-staining. While both contain differing information, we have confirmed via personal communication with pathologists specializing in breast cancer [7] that H&E includes sufficient information in order to be able to create p63 stained images from it.

As such, and because of the often significant differences between the two tissue stains as seen in Figure 1, we have not registered the image pairs and instead use a model architecture based on CycleGAN to achieve reliable conversion without the need for aligned paired samples.

### 3.1. Properties of Data

The data consist of 36 pairs of H&E and p63 slides from various patients, with a wide range of possible resolutions for each tissue sample. The resolutions range from as little as 20,000 by 36,000 pixels to upwards of 120,000 by 96,000 pixels saved in the Olympus VSI [27] format, requiring specialized libraries to open and process the images.

The images are saved in ordinary RGB format with various scales of the slide available, reducing the quality with each step.

The background is predominantly white with non-uniform intensity, which necessitates separating actual tissue from the background.

### 3.2. Preprocessing

Due to the nature of the file format and the necessity of specialized libraries to open and process the images, we split the slides ahead of time into 1024 by 1024 pixel non-overlapping regions, which are then down-scaled to 256 by 256 pixels for training.

We use entropy to detect whether a region contains a sufficient amount of tissue compared to the background or whether the tissue is sufficiently differentiated across the region to have a reasonable information density. Regions that do not match a base minimal entropy are excluded.

Further, we convert the images into the L*a*b* color space as it was shown that convolutional networks can handle it better, and it is more linearly separable. Especially in our case, where p63 images contain primarily brown cells and blue tissue, the L*a*b* color space allows us to separate those easily. The same applies to H&E images, which range from light pink to dark purple and are also linearly separable in the L*a*b* color space.

### 3.3. Post-Processing

We calculated the mean and standard deviation of the luminance and chrominance channels of the data as well as the minimum and maximum values. We utilize tanh(x) activation function on the end of the model, which is particularly compatible with the L*a*b* color space, and normalize the image back into the proper range by utilizing these maximal and minimal values, further aided by a trainable luminance multiplier parameter to help the model achieve the necessary color accuracy in the image.

Furthermore, we implemented a differentiable joint-bilateral guided blur [28] based on the library kornia implementation [29] which is a part of the gradient computation. This smoothes out some artifacts and improves the image quality. We followed this with an unsharp mask to return clarity to the image after the blurring.

## 4. Methods

The xAI-CycleGAN architecture was further improved and stabilized by introducing attention-based skip-connection merging at the lowest two encoder–decoder layers where the most important features are located.

Additionally, we added a modified context-preserving loss mechanism that significantly more strictly monitors the preservation of structural and semantic information at the latent layers that are independent of color information.

### 4.1. Context-Preserving Loss

The context-preserving loss ensures that the encoded representation of an image is the same in both generators in a given direction based on the domain. This serves a similar function to classification networks present alongside CycleGAN architectures for domain transformation in histology but without relying on the identification of any particular classes.

The loss is computed on the basis of Huber loss, which works well to penalize significant differences more effectively while using a milder gradient for small ones.

Given *A* as original domain A image, $A^{\prime }$ as fake domain A image, $G_{enc}^{AB}$ as generator encoder for the A to B direction, $G_{conv}^{AB}$ as the generator latent transformation module for the A to B direction (which would be immediately followed by the decoder) and all alterations for domain B and the other direction, respectively, the piece-wise context loss is defined as:(1)$$C_{\alpha }=H\left(G_{enc}^{AB}\left(A\right),G_{conv}^{BA}\left(B^{\prime }\right)\right)$$
(2)$$C_{\beta }=H\left(G_{conv}^{AB}\left(A\right),G_{enc}^{BA}\left(B^{\prime }\right)\right)$$
(3)$$C_{\gamma }=H\left(G_{enc}^{BA}\left(B\right),G_{conv}^{AB}\left(A^{\prime }\right)\right)$$
(4)$$C_{\delta }=H\left(G_{conv}^{BA}\left(B\right),G_{enc}^{AB}\left(A^{\prime }\right)\right)$$

Therefore, the final context-preserving loss can be defined as:(5)$$L_{context}=\left(\frac{C_{\alpha }+C_{\beta }}{2}\right)\cdot \gamma_{A}+\left(\frac{C_{\gamma }+C_{\delta }}{2}\right)\cdot \gamma_{B}$$
where $\gamma_{A}$ and $\gamma_{B}$ are hyper-parameters, which we have set to 5 for both. A visual representation of $C_{\alpha }+C_{\beta }$ is presented in Figure 2.

### 4.2. Convolutional Attention Skip Connection Merging

We use convolutional attention based on self-attentive GANs [30,31] to merge the two bottom-most encoded representations. Combined with the context loss, this ensures that the generator learns to understand the structure and information contained within the tissue rather than just using the skip connection to cheat the transformation process. However, we still preserve an encoded skip connection adjusted by a learnable parameter as in the original paper.

We use a large reduction number of 64 and 32, respectively, which reduces the number of channels to 4 in both cases for the attention mechanism, which makes it significantly more efficient on the memory, only requiring 12 GB of VRAM for the whole model including an image passed through it during training.

This could be further reduced by methods such as sliding window attention or other modern methods of optimizing the attention mechanism.

### 4.3. Semantically Slightly Significant Editing

We have adapted the SeFa [22] algorithm and improved it further over the previous attempt at employing editable output in xAI-CycleGAN [9] by changing how the extracted eigenvalues are applied to the weights of the interpretable latent variable layer, which has features distilled into it from training due to explainability powered training.

Like SeFa, we extract semantically significant directions using eigenvalues from the weight matrix. However, like in our previous work, we only utilize a single weight matrix, unlike several merged together in SeFa, while still achieving semantically interpretable editing, now including a choice of several ’significance’ steps where each further step uses a less significant eigenvector for editing the output, thus, less semantically interpretable or meaningful.

We obtain the normalized weight matrix from the interpretable latent variable convolutional layer, labeled *W*:(6)$$W_{norm}=\frac{W}{{\left|W\right|}_{2}}$$

From this, we obtain the semantically significant directions by extracting the eigenvalues of the normalized matrix multiplied by a transpose of itself:(7)$$E=eig(W_{norm}W_{norm}^{T})$$

Using an adjustable parameter *r* we then obtain the top *r* significant matrices from *E*:(8)V=top(E,r)

After that, we compute the feature map *X*, which are the activations just past the convolutional layer of the interpretable latent variable from which the weight matrix was obtained. For every significant extracted matrix, adjusted by a modifier α, we apply matrix multiplication with the feature map *X* and sum the results before adding the sum to the original feature map:(9)$$X_{mod}=X+\sum_{i=0}^{r}\left(X⊗\left(V_{i}\cdot \alpha_{i}\right)\right)$$

We then pass the modified feature map $X_{mod}$ through the rest of the generator to obtain our edited image.

## 5. Results

Here, we would like to present the improved editing capabilities as well as a simple quantitative evaluation and comparison with regard to our previous work. We demonstrate preserved editing capabilities as well as higher predictability of editing.

### 5.1. Training Evaluation

To compare the results, we use Fréchet Inception Distance (FID) [32], which is traditionally used in generative model scenarios where other metrics are often unviable. We compare the original xAI-CycleGAN with our enhanced version on the same test data, averaging the FID of the H&E and p63 domains, as seen in Figure 3.

Additionally, with these modifications, we also solved the issues of the explainability enhanced training demonstrated in Figure 4, which would start to produce artifacts later during training as seen in Figure 4a in order to trick the discriminator and achieve a lower loss artificially.

This improves over both our previous work [9] and xAI-CycleGAN, and is another significant step towards reliable conversion between histological stains and without artifacts as seen in Figure 4b. The editing capability remains preserved as well and allows us to produce converted images even more closely and accurately, though not yet in an automated fashion.

### 5.2. Histopathologist Evaluation

We have constructed a short survey where three histopathologists were presented with eight pairs of images, one fake p63 sample generated from H&E and a ground truth p63 sample from the same region. Naturally, differences exist between these images and they cannot be the same, but they are relatively similar in tissue structure and with a matching location in a paired H&E and p63 tissue slide. The results are presented in Table 1.

Interestingly, the correct images were identified almost all the time, along with a decreased realism score compared to our previous study [9]. However, we allowed the pathologists to add a summary commentary on what helped them identify the correct images and an overall summary of the images presented.

This reveals that most of the time the correct images were identified based on the perceived sharpness of contours and deeper and more pronounced shades of blue, but we also learn that the accuracy was otherwise very good and while most of the time the image was identified correctly, the pathologists had big doubts on whether their choice was correct.

There remain several inaccuracies in the re-staining, but overall the accuracy has improved significantly, but new elements were introduced that made it easier to identify the generated image in direct comparison. However, we do not necessarily view this as a downside, since the overall accuracy has increased.

### 5.3. Editing Capabilities and Tool Modifications

The images of the end result still remain high quality and slightly inaccurate to the actual ground truth, which is, however, solvable by editing capabilities as demonstrated in our previous work, seen in Figure 5.

These editing capabilities are preserved and, in fact, improved with the new algorithm. Previous various modulation ranges were adjusted by a single modifier, and the results were hard to control and predict and were not interpolable.

With the new algorithm, we are not only able to still edit images to match ground truth for most cases better, but we are also able to demonstrate interpolability and predictable editing.

Each eigenmatrix now has its own modifier α based on how many top K eigenmatrices we have selected, which modify each independently before being added to the activations.

For this, we have also modified the Streamlit [33] app (ver. 1.29.0) to reflect these changes and allow editing of the output with these new per-eigenmatrix modifiers.

We chose to demonstrate the editing on an image that is not supposed to contain myoepithelial cells to highlight a potential downside of the editing approach. In Figure 6a, we can see an unedited H&E image of tissue where there are supposed to be no myoepithelial cells present, followed by an edited image in Figure 6b.

This shows that the method is also capable of producing images that are inaccurate or misleading. Due to the technique used and exiting adversarial attacks, it may be possible to include imperceptible noise to the input image, which would achieve the given edited output in Figure 6b without editing.

Various state-of-the-art methods for mitigating this issue use diffusion models, adversarial augmentation and others. Still, we have yet to find practical techniques for our modified CycleGAN model, and it is thus an area of potential further research.

You can test the model and editing capabilities with this demo image yourself using the supplied code on our GitHub repository (https://github.com/slobodaapl/editable-stain-xaicyclegan2, accessed on 8 January 2024), as it is pre-configured to run with the model checkpoint provided in its current form.

In Figure 7, we demonstrate the semi-interpretable stable editing capabilities of the method. We have chosen extreme ranges of α, which we otherwise limit to the range (−3.0;3.0) as we observed the most stable results in this range.

The extreme ranges of α demonstrate various artifacts begin to form in primarily high positive α values when multiple eigenmatrices are selected. Otherwise, we can observe specific aspects of the image changing in relatively predictable ways as compared to the previous attempt in our last paper [9].

## 6. Discussion

In this paper, we have presented advancements to the editable xAI-CycleGAN for histopathological stains, with improvements to the editing capabilities and context-preserving loss, as well as the quality of output proven by FID. Here, we briefly discuss these advancements, their implications and their potential for future research.

### 6.1. Advancements in Context Preservation and Attention Mechanism Inclusion

The enhanced xAI-CycleGAN model demonstrates a significant leap in the realm of histological stain transformation. By incorporating a more robust context-preserving loss, along with modern methods of convolutional cross-attention, we have achieved a notable improvement in the model’s capacity to comprehend and maintain the context of images at the latent level. This approach is particularly evident in the merging of latent representation with the skip connection from the encoder in the decoder step. Additionally, we made the code publicly available on GitHub (https://github.com/slobodaapl/editable-stain-xaicyclegan2, accessed on 8 January 2024).

### 6.2. The Potential of Transformers in Cycle-Consistent Generative Networks

Our results underscore the substantial potential of transformers and attention mechanisms in further enhancing cycle-consistent generative networks. While there are some explorations in the literature involving partially transformer-based approaches, our work suggests that fully transformer-based cycle-consistent generative networks could represent a significant advancement in this field. These networks, equipped with self-attention maps from fully transformer-based discriminators and generators, could potentially prevent the need for a second discriminator or even the masking element in xAI-CycleGAN. This would mark a notable simplification of the architecture, potentially leading to more efficient training and better results.

### 6.3. Challenges and Future Directions

A significant challenge that remains is the computational complexity and hardware requirements associated with fully transformer-based models. The implementation of attention-based mechanisms, although promising, demands substantial computational resources. Future work should explore methods to optimize these mechanisms, making them more computationally efficient and less resource-intensive.

We see great potential in creating a fully transformer-based CycleGAN architecture with the inclusion of explainability-based training due to the utilization of cross-attention and attention maps from the discriminator.

### 6.4. Improving Editing Capabilities

Our ongoing efforts will focus on enhancing the model’s explainability-based training and editing capabilities. We aim to improve the model’s interpretability and the quality of generated images by utilizing self-attention maps from fully transformer-based architectures. This enhancement is expected to lead to more realistic and accurate conversions of histological stains, facilitating better diagnosis and research in pathology.

## 7. Conclusions

In summary, our enhanced xAI-CycleGAN model represents a positive step forward in the field of histological stain transformation, albeit our approach is not yet comparable to original digital images of p63 stained tissue and cannot yet be used for diagnosis. The introduction of a more effective context-preserving loss and the integration of convolutional cross-attention mechanisms have, however, markedly improved the model’s performance. Looking forward, fully transformer-based cycle-consistent generative networks hold great promise for further advancements in this domain, although they bring challenges that must be carefully addressed. The potential improvements in explainability, editing capabilities and overall efficiency of these models make them a promising avenue for future research and development.

## Figures and Tables

**Figure 1 jimaging-10-00032-f001:**
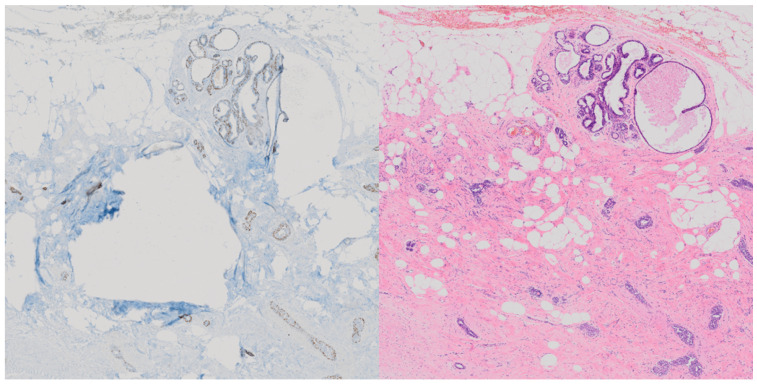
Demonstration of significant differences in tissue in paired and aligned p63 stained tissue (**left**) compared with its H&E counterpart (**right**) [9].

**Figure 2 jimaging-10-00032-f002:**
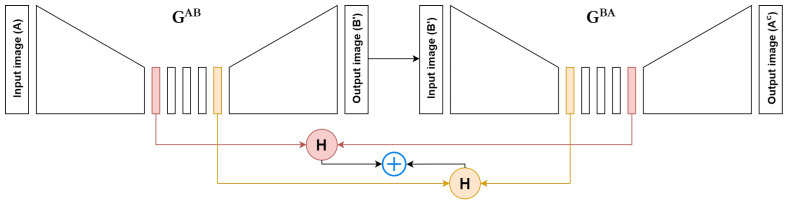
Demonstration of the context loss computation in one direction. $G^{AB}$ represents the H&E to p63 conversion and $G^{BA}$ represents the p63 to H&E conversion. $B^{\prime }$ is the converted (fake) p63 image and $A^{c}$ is the cycled H&E image. *H* represents Huber Loss.

**Figure 3 jimaging-10-00032-f003:**
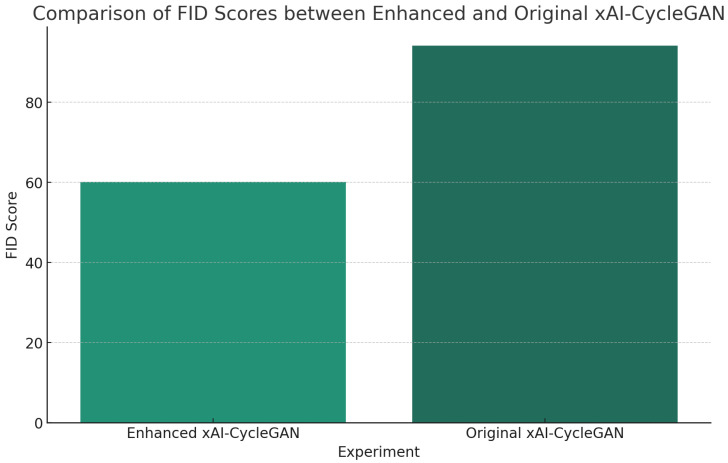
Results show that the FID of the enhanced xAI-CycleGAN is significantly lower than that of the original, demonstrating significant improvement over the previous method.

**Figure 4 jimaging-10-00032-f004:**
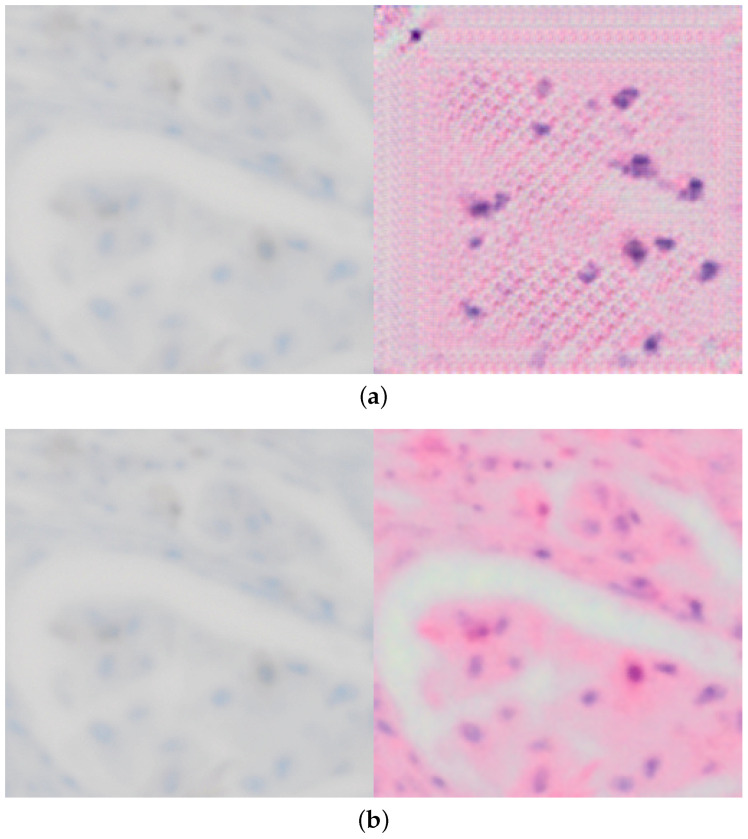
Comparing conversion from p63 to H&E for both original and improved xAI-CycleGAN at the same point in training, demonstrating that the issue with corruption/artifacts in conversion is solved. (**a**) Demonstration of the corrupted conversion present in original xAI-CycleGAN due to explainability-driven training. (**b**) Demonstration of a clean proper conversion with our enhanced model, using the same test image, at the same point during training (the model has seen the same amount of training samples).

**Figure 5 jimaging-10-00032-f005:**
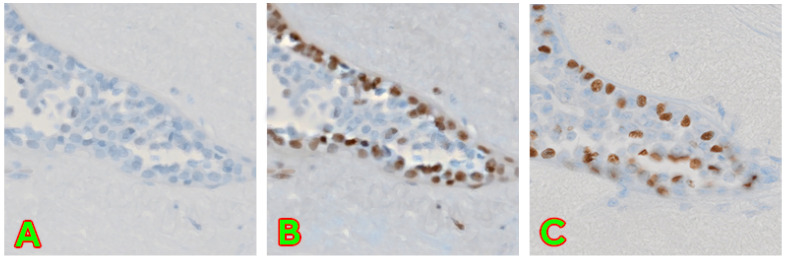
A demonstration of successful editing capabilities. (**A**) contains image converted from H&E to p63 without any modifications applied. (**B**) contains an image with modifications applied to best match the real image. (**C**) contains the unmodified original p63 image of the same region. The same base image as in our previous work [9] has been used.

**Figure 6 jimaging-10-00032-f006:**
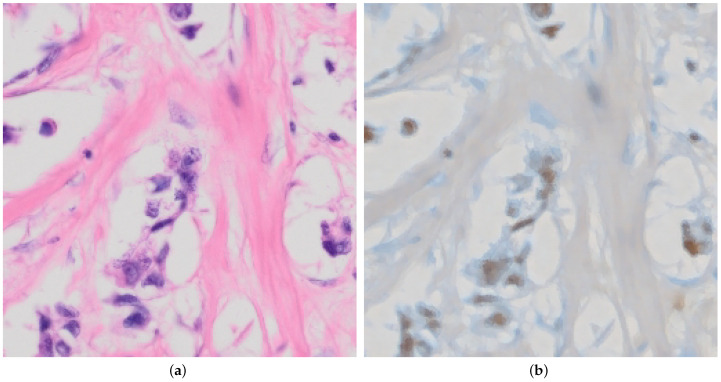
Demonstration of conversion by our model from H&E to p63 where no myoepithelial cells are present. (**a**) Raw unedited H&E that has no myoepithelial cells present. (**b**) Edited p63 image after correct transformation to include brown coloring by p63 for myoepithelial cells, which is not meant to be present in the image. For this image, we used r=3 with α=[−2,−1.6,0.2] for each vector, respectively.

**Figure 7 jimaging-10-00032-f007:**
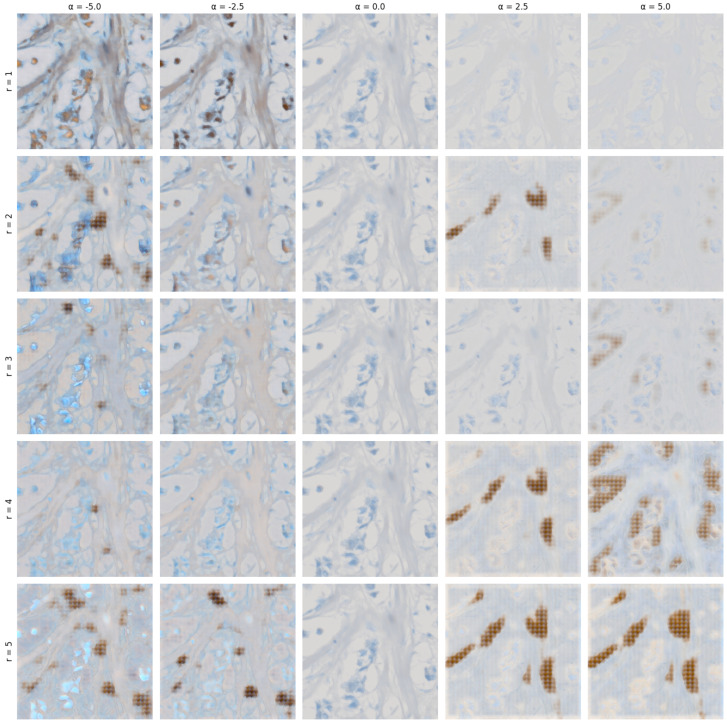
A grid of edited images with varying *r* from 1 to 5, and α values for columns ranging from −5.0 to 5.0 with a step size of 2.5 (applied to all matrices *V* defined in Equation (Equation 8)).

**Table 1 jimaging-10-00032-t001:** A total of eight image pairs were presented to three histopathologists; a generated image and a ground truth matched image. Each histopathologist was asked to guess the real image and rate how realistic the generated image looks from 1 to 6. Incorrect guesses were assigned a score of 6 for realism of the generated image. The last row represents the summed total of the correctly and incorrectly identified images, and the average realism rating for the last column. In total, 87.50% of the images were identified correctly.

	Correctly Identified	Incorrectly Identified	Average Realism Rating
Pair 1	3	0	4
Pair 2	3	0	2.67
Pair 3	2	1	4.67
Pair 4	3	0	2.34
Pair 5	3	0	3.67
Pair 6	2	1	4.67
Pair 7	1	2	5
Pair 8	2	1	5.67
Total/Average	21	3	4.08 (max 6)

## Data Availability

The data presented in this study are available on request from the corresponding author. The data are not publicly available due to being internal histopathological data belonging to the Institute for Clinical and Experimental Medicine in Prague purchased privately for our use in this study.

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
