# Peer review of "Attention-Enhanced Unpaired xAI-GANs for Transformation of Histological Stain Images"

_2313-433X, 2024, doi:10.3390/jimaging10020032_

Round 1

Reviewer 1 Report

Comments and Suggestions for Authors

The authors present a description of digital transformation of histological images of slides stained with conventional haematoxylin and eosin (H&E) into simulated images stained with the immunohistochemical stain p63 (and note that most publications write p63, rather than P63). They justify this submission because of some improvements over their recent publication (Sloboda et al, 2023), but these two publications have three Figures in common, which I would regard as poor publication practice, given that the figure legends in the current publication make no attempt to indicate that the images have already been published. I find no particular evidence that improved transformation would improve histological interpretation, and there is no explanation why they tested their improvement on a dataset from the rare lesion of metaplastic breast cancer; there are many more common lesions where understanding the distribution of p63 is valuable.

Their justification of developing reciprocal digital transformation between H&E and p63 is that p63 is expensive and damages tissue. I have worked with p63 since its introduction into diagnostic practice, and I have seen no evidence of a predilection to damage tissue; indeed, the histomorphometry is very easily interpreted with p63 in almost all cases. P63 is a widely used reagent, and its expense and safety are comparable to many other routine immunohistochemical stains. The authors need to provide references to support the contentions in lines 31 to 35. Further, their justification for using these transformations include interviews with pathologists (lines 146 to 148), and it would be useful to have more details of these interviews. For instance, were they formal interviews based on a pre-written questionnaire? Or something more akin to opportunistic corridor conversations? And without actual summative evaluations from histopathologists (lines 258 to 261), the validity of their claims remain in doubt. The gold standard for diagnostic utility will be blinded interpretations of digitally transformed slides compared to digital images of real slides.

The references are generally adequately presented, though reference 7 lack a publisher ( it was “2017 IEEE International Conference on Computer Vision (ICCV)”. Some titles are littered with capital letters, and others are not; consistency is generally preferred. Reference 25 appears to have an incomplete website address, and reference 26 has no website address (despite an accession date implying that there should be one)

Sloboda, T.; Hudec, L.; Benešová, W. Editable Stain Transformation Of Histological Images Using Unpaired GANs. arXiv preprint 384 arXiv:2312.03647 2023

Comments on the Quality of English Language

Generally good, with no critical flaws.

Author Response

Dear reviewer,

Thank you for your insightful comments and suggestions. We have carefully considered each point and made the following revisions to our manuscript:

Regarding the Reuse of Figures:
We acknowledge the oversight in reusing figures from our previous publication. We have now replaced the mistakenly reused Figure 5 with the updated one, which better aligns with the current study. Additionally, we have revised the Figure 2 representing the context-preserving loss to more closely mirror the described equations, updated to the improved version. We retained one image from our previous work but have now clearly labeled its reuse in the Figure 1 caption to maintain transparency.

Claim about p63 Damaging Tissue:
We appreciate your expertise on p63 and have revised our claim of tissue damage after consultation with experts in the field. Lines 31-35 contain revised information, highlighting primarily the significantly higher cost as compared to H&E, as well as increased difficulty of preparation and time commitment. This information has also been updated in lines 147-149 of the revised manuscript.

Choice of Dataset:
The dataset from the Institute for Clinical and Experimental Medicine focusing on this particular type of breast cancer was chosen due to our existing rights to use it and our familiarity with its specific pre-processing requirements. We acknowledge that there are more common lesions where understanding the distribution of p63 is valuable. However, changing to a different dataset would require significant effort and is beyond the scope of this study. Our objective was not to demonstrate performance across various datasets but to focus on the improvements in our digital transformation technique. The chosen dataset reflects our ongoing work and expertise in this particular tissue and cancer type. To this end we have provided a short explanation in the revised manuscript on lines 133-136 as to the choice of the dataset and the focus on computer vision and deep learning methods, rather than the focus on creating a diagnostic tool.

Histopathologist Evaluations:
We have managed to obtain histopathologist evaluation which is now available in the manuscript.

Adjustments to References:
We have revised the references for consistency and accuracy. The missing publisher information for reference 7 has been added, and the incomplete website address in reference 25 has been corrected. We apologize for the oversight regarding the missing link in reference 26, which was an editing error post-submission. This has now been rectified.

We appreciate your feedback, which has helped us refine our manuscript and clarify our study's aims and limitations. We hope that these revisions adequately address your concerns.

Sincerely,
Tibor Sloboda

Reviewer 2 Report

Comments and Suggestions for Authors

In this manuscript, the authors describe the "Attention-enhanced Unpaired xAI-GANs for Transformation of Histological Stain Images". While this manuscript is generally well written, it would be helpful if the authors address the following minor concerns below.

1) In line 240, there is no need to use contractions. The sentence should be "Here we would like to present.....". The authors should consider revising this.

2) To avoid unnecessary distraction, the two subfigures in figure 4 should be on the same page to facilitate proper comparison.

3) The two images in figure 4 should be labelled differently. Currently they are both labelled as 4a which is not only confusing, but also inconsistent with the comparison provided in figure 4 description. The authors should correct this.

In summary, this manuscript could be of benefit to its target readers if the above concerns are addressed.

Comments on the Quality of English Language

The quality of the English language is good.

Author Response

Dear Reviewer,

Thank you for your constructive feedback on our manuscript. We value your insights and have taken immediate steps to address the points you raised.

1. We sincerely apologize for the unintentional inclusion of a contraction in line 240. It was certainly not our intention, as we had diligently endeavored to eliminate all contractions in the initial draft. This oversight has now been rectified in the revised version.

2. Regarding the two subfigures in figure 4, we acknowledge that an editing error resulted in their separation across pages. This has been corrected, and both subfigures are now presented on the same page for easier comparison.

3. The labeling issue in figure 4 was also a result of an editing mistake. We have amended this, ensuring each image in figure 4 is uniquely and correctly labeled.

We appreciate your attention to these details and hope that our revisions meet your expectations. These will be included in our upcoming manuscript submission.

Sincerely,
Tibor Sloboda

Reviewer 3 Report

Comments and Suggestions for Authors

The authors describe a method to improve histological staining to improve detection of metastatic breast cancer. The methods of analysis are clearly described and the data supports their conclusions. I recommend publication after major revisions due the missing evaluation of an expert pathologist. In addition, I have a couple minor issues with Figures 4 and 6. 

Figure 4: Needs to be edited to reflect what is in the description.

Figure 6: I am a little confused about the relevance of this figure and also the figure itself that shows a place to load a file. Please consider editing or removing.

Author Response

Dear Reviewer,

Thank you for your thoughtful review and recommendation for our manuscript. We appreciate your recognition of the clarity and support provided by our analysis in improving histological staining for the detection of metastatic breast cancer. We understand your concerns regarding the involvement of an expert pathologist and the presentation of Figures 4 and 6.

Regarding the evaluation by an expert pathologist, we have now obtained evaluation of three expert pathologists which is now present in the manuscript.

For Figure 4, we will ensure it accurately reflects the content described in its caption. We apologize for any discrepancy and appreciate your attention to detail in highlighting this.

Regarding Figure 6, we understand your confusion about its relevance and presentation. After re-evaluating it with the co-authors we have decided to remove this figure altogether.

We thank you for your attention to detail and the review of our manuscript on behalf of all authors.

Sincerely,
Tibor Sloboda

Round 2

Reviewer 1 Report

Comments and Suggestions for Authors

The authors are to be congratulated on the changes they've made to the manuscript, and they have made an appropriate response to my constructive criticisms. I think that their new reference 7 would be better referenced as a personal communication in the text, but I don't feel strongly about it.

Comments on the Quality of English Language

There remain idiosyncrasies of the use of words and phrases, but these can be dealt with during subediting.

Author Response

Dear Reviewer,

Thank you for your constructive feedback on our manuscript. We have taken your suggestion into account and have updated line 151 which mentions it to a personal communication. Additionally, we have modified the corresponding reference accordingly to reflect this change. We appreciate your guidance and support in enhancing our work.

Best regards,
Tibor Sloboda

Reviewer 3 Report

Comments and Suggestions for Authors

The authors have addressed my concerns and the manuscript should be accepted. 

Author Response

Dear Reviewer,

Thank you for your positive evaluation and the recommendation for acceptance of our manuscript. We are grateful for your insightful feedback throughout the review process, which has contributed to the improvement of our work.

Best regards,
Tibor Sloboda